# Hyper-ICL: Attention Calibration with Hyperbolic Anchor Distillation for Multimodal In-Context Learning

Niloufar Alipour Talemi [1]  Hossein Kashiani [1]  Fatemeh Afghah [1]

## Abstract

Multimodal In-Context Learning (ICL) has emerged as a practical inference paradigm for Multimodal Large Language Models, where a small set of interleaved image-text In-Context Demonstrations (ICDs) conditions the model to solve new tasks. Despite its flexibility, multimodal ICL incurs high inference latency and suffers from instability due to sensitivity to demonstration formatting, ordering, and content. To address these limitations, we propose Hyper-ICL, a lightweight, training-based framework for demonstration-free multimodal ICL that reconstructs demonstration effects directly without requiring ICDs at inference time. Hyper-ICL learns a parameter-efficient low-rank logit-level adapter that calibrates attention distributions to better match demonstration-induced attention redistribution. To capture how demonstration influence varies across queries, we introduce a query-adaptive modulation mechanism that adaptively controls intervention strength at token level across layers and heads based on the current query. Finally, we propose a layer-wise hyperbolic anchor distillation loss that aligns intermediate student features to a demonstration-conditioned teacher via Lorentz geodesic distance. This loss encourages the student to reconstruct the demonstration–query relationships induced by ICDs. Extensive experiments across six different multimodal benchmarks (including VQAv2, OK-VQA, and COCO Caption) demonstrate that Hyper-ICL consistently improves accuracy and stability over vanilla ICL and existing state-of-the-art methods.

[1]Holcombe Department of Electrical and Computer Engineering, Clemson University, Clemson, United States. Correspondence to: Niloufar Alipour Talemi <nalipou@clemson.edu>.

*Proceedings of the $43^{rd}$ International Conference on Machine Learning*, Seoul, South Korea. PMLR 306, 2026. Copyright 2026 by the author(s).

## 1. Introduction

Multimodal Large Language Models (MLLMs) (Zhou et al., 2022; Li et al., 2023; Liu et al., 2023) have recently become a strong paradigm for unified vision-language understanding and generation, enabling a single model to solve diverse tasks such as Visual Question Answering (VQA) (Shao et al., 2023; Goyal et al., 2017), image captioning (Chen et al., 2015) and visual reasoning (Xu et al., 2026) with minimal task-specific adaptation. A particularly attractive adaptation mechanism is In-Context Learning (ICL) (Liu et al., 2022), where the model is steered at inference time by prefixing a small set of In-Context Demonstrations (ICDs) into the prompt (Dong et al., 2024). Compared to parameter updating, ICL offers rapid and flexible adaptation, making it a practical interface for deploying MLLMs across diverse downstream applications.

Despite recent progress in ICL, multimodal ICL still faces several practical and fundamental limitations. Compared with text-only ICL in Large Language Models (LLMs), MLLMs require interleaved image–text demonstrations that substantially increase the effective input length, leading to higher inference latency. Beyond this token overhead, recent studies (Li et al., 2024b; Qin et al., 2024) show that MLLMs can overfit to superficial ICD cues, leading to brittle behavior under formatting and ordering changes (Sun et al., 2025; Zhang et al., 2023; Baldassini et al., 2024). For instance, in VQA, MLLM may mimic the answer format observed in the demonstrations rather than learning the underlying input–output mapping, in contrast to behavior more commonly seen in language-only QA. These instabilities are exacerbated in multimodal contexts, where the inherent complexity of aligning cross-modal features introduces additional layers of variance.

Recent work has shifted toward vector-based ICL to mitigate these overheads and reproduce in-context behaviors without long, token-heavy prompts. Rather than conditioning on raw ICDs, these methods extract a compact task representation from demonstrations and inject it into the model as an additive shift in the hidden activations, reframing ICL as the induction of an internal task representation by the input context (Brown et al., 2020; Todd et al., 2023). Compressing multiple demonstrations into a single In-Context Vector (ICV) shortens the inference-time prompt and reduces re-

liance on carefully selecting and ordering ICDs. However, heuristic ICV extraction often fails on complex multimodal tasks like VQA, which require richer cross-modal reasoning and tighter visual-language alignment than language-only scenarios.

To address these challenges, a recent study, LIVE (Peng et al., 2024), proposes a training-based approach that learns a rich shift vector from a large supporting set. LIVE distills task-relevant information from many randomly sampled ICD sets into layer-wise ICVs, achieving superior performance over heuristic methods. However, it remains limited in modeling the complex interactions inherent in multimodal ICL. As shown by both empirical results and theoretical analysis in Section 3.1, not only layers but also attention heads play distinct roles in processing demonstrations, which is an aspect LIVE overlooks. Moreover, to fully exploit cross-domain ICD patterns, such information should be injected into attention logits rather than final-layer hidden states. Lastly, LIVE applies a uniform adaptation across inputs, which can degrade performance when queries require different degrees of adjustment.

Building on these insights, we propose Hyper-ICL, a lightweight, training-based framework that decomposes and reconstructs multimodal in-context effects directly within the attention mechanism (shown in Figure 1). Instead of injecting a uniform hidden-state shift, Hyper-ICL learns a lightweight low-rank logit-level adapter that steers where the model attends in a way that better matches the attention patterns induced by ICDs. To accommodate the query-adaptive nature of ICL, we further introduce a query-adaptive modulation mechanism that dynamically controls intervention strength across layers and heads, ensuring query-adaptive calibration while suppressing unnecessary interference.

Finally, we propose a layer-wise hyperbolic anchor distillation loss that aligns intermediate student representations with a demonstration-conditioned teacher via hyperbolic geodesic distance, allowing the student to recover both high-level guidance and fine-grained contextual refinements even without access to demonstrations at inference time. Hyperbolic geometry is particularly well suited to this setting, as multimodal ICL induces a dense set of demonstration–query relationships that must remain coherent across intermediate layers. Prior work motivates negative curvature as offering effective representational capacity and mitigate distortion in low-dimensional embeddings (Fish & Bowden, 2025; Ibrahimi et al., 2024). This helps preserve relative similarity ordering when many interactions are embedded in a low-dimensional embedding (Law et al., 2019). In our work, the teacher encodes these demonstration-conditioned relations, whereas the student observes only the query. Aligning their intermediate states with Lorentz geodesic distance, therefore, encourages the student to match the teacher's multi-scale relational geometry, rather than relying on Euclidean proximity that can permute relative distance structure (Law

et al., 2019; Poppi et al., 2025).

In summary, our main contributions are:

- We introduce **Hyper-ICL**, an efficient multimodal ICL framework that decomposes demonstration effects within self-attention and proposes a logit-level attention intervention that directly calibrates attention distributions rather than approximating demonstration-induced shifts only at the output level.

- We propose a layer-wise hyperbolic anchor distillation loss that aligns intermediate student features to a demonstration-conditioned teacher via Lorentz geodesic distance, helping preserve relative similarity ordering under dense demonstration–query relationships and enabling demonstration-free inference.

- Experiments on two large-scale MLLMs across six widely used, challenging benchmarks (including VQAv2 (Goyal et al., 2017), OK-VQA (Marino et al., 2019), and COCO Caption (Chen et al., 2015)) show that Hyper-ICL consistently improves performance and stability over direct ICD prompting, vector-based ICV baselines, and training-based alternatives.

## 2. Related Work

### 2.1. Multimodal Large Language Models

With advances in LLMs, MLLMs have emerged as a unified framework for joint vision–language understanding and generation (Liu et al., 2023; Alipour Talemi et al., 2025; Chen et al., 2024). A widely adopted formulation augments a pretrained LLM with a vision encoder and a lightweight alignment module, such as a projection layer or a query-based transformer, to map visual features into the LLM embedding space for multimodal reasoning (Liu et al., 2023). Idefics-9b (Laurençon et al., 2023) follows a cross-attention based architecture, where language features attend to image representations through dedicated attention blocks for multimodal integration. In contrast, Idefics2-8b (Laurençon et al., 2024) adopts a fully autoregressive design by converting visual features into token-like embeddings and processing them jointly with text via self-attention. A widely adopted strategy for adapting MLLMs to new tasks is ICL, where task behavior is induced by conditioning on a few multimodal demonstrations at inference. Despite its simplicity, multimodal ICL is less stable than language-only ICL, as cross-modal interactions increase sensitivity to demonstration choice, order, and formatting (Ma et al., 2025).

### 2.2. In-Context Learning

ICL enables rapid task adaptation by conditioning on demonstrations at inference time, without updating model parameters. However, in multimodal ICL, each image introduces substantial token overhead, making high-shot prompting expensive and highly sensitive to demonstration choice and formatting (Doveh et al., 2024; Li et al., 2026; Baldassini et al., 2024; Li et al., 2025c). To reduce this cost, exist-

ing approaches develop efficient ICL mechanisms such as vector-based compression that distills demonstration effects into compact ICVs injected into intermediate activations, as well as lightweight activation-shift strategies that approximate ICL through additive shifts estimated from attention statistics or layerwise representations (Brown et al., 2020; Todd et al., 2023). LIVE (Peng et al., 2024) follows a related direction by learning task-specific, layer-wise ICVs via distillation from demonstration-conditioned outputs, improving VQA performance while lowering inference overhead, but it still fails to precisely recover the true, query-adaptive demonstration shifts. Motivated by these limitations, we propose Hyper-ICL to faithfully decompose the effects of ICDs for effective multimodal ICL.

### 2.3. Learning in Hyperbolic Space

Hyperbolic geometry has emerged as an effective alternative to Euclidean representation learning when data exhibit latent hierarchies, power-law degree distributions, or tree-like structures. Early work such as Poincaré embeddings showed that spaces with constant negative curvature can represent hierarchical relations with lower distortion and higher capacity than Euclidean spaces of the same dimensionality (Nickel & Kiela, 2017). Follow-up studies demonstrated that the Lorentz model improves numerical stability and optimization efficiency for large-scale taxonomy learning while preserving the same geometric advantages (Nickel & Kiela, 2018). Building on this, hyperbolic deep learning extends neural operations to Riemannian manifolds via exponential and logarithmic maps, enabling hyperbolic analogues of linear layers, recurrent units, and classification heads (Ganea et al., 2018). Recently, hyperbolic objectives have been explored in vision and multimodal learning to encode compositional scene-object hierarchies and structured vision-language alignment (Ge et al., 2023). In this work, we adopt the Lorentz model (Law et al., 2019) as a geometry-aware alignment space, using a layer-wise hyperbolic anchor distillation loss to match intermediate student representations to teacher hidden states via geodesic distance, yielding a structure-preserving regularizer that maintains latent demonstration–query relationships.

## 3. Proposed Method

### 3.1. Motivation and Problem Setup

ICL enables LLMs and MLLMs to generalize to new tasks by conditioning on a small set of ICDs provided directly in the input. We define the prompt context as $C = \{X_D, X\}$, where $X_D = \{X_1, X_2, \ldots, X_m\} \in \mathbb{R}^{T_D \times d}$ denotes the concatenation of $m$ ICDs, and $X \in \mathbb{R}^{T \times d}$ is the query input. Here, $T_D$ and $T$ represent the number of tokens in $X_D$ and $X$, respectively, and $d$ is the embedding dimension. Multi-head self-attention applies the self-attention (SA) mechanism across $N_h$ heads. Each head is parameterized by projection matrices $W_k, W_q, W_v \in \mathbb{R}^{d \times d_h}$, which

map $C$ to keys $K_C$, queries $Q_C$, and values $V_C$. Typically, $d_h$ is set to $d/N_h$ so that each head operates in a lower-dimensional subspace, reducing parameter usage while preserving expressivity. For a given head, the key mapping is defined as follows:

$$K_C = C W_k = \begin{bmatrix} X_D \\ X \end{bmatrix} W_k = \begin{bmatrix} K_D \\ K \end{bmatrix}. \qquad (1)$$

Similarly, we compute the corresponding $Q_D, Q$, and $V_D, V$ using $W_q$ and $W_v$, respectively. For a query vector $q \in Q$, the single-head self-attention computation is given by (for clarity, we present the formulation for a single head):

$$
\begin{aligned}
\mathrm{SA}\left(q, \begin{bmatrix} K_D \\ K \end{bmatrix}, \begin{bmatrix} V_D \\ V \end{bmatrix}\right) &= \mathrm{softmax}\left([qK_D^\top, qK^\top]\right) \begin{bmatrix} V_D \\ V \end{bmatrix} \\
&= \left[\frac{\exp(qK_D^\top)}{Z_1 + Z_2}, \frac{\exp(qK^\top)}{Z_1 + Z_2}\right] \begin{bmatrix} V_D \\ V \end{bmatrix} \\
&= \frac{Z_2}{Z_1 + Z_2} \frac{\exp(qK^\top)}{Z_2} V + \frac{Z_1}{Z_1 + Z_2} \frac{\exp(qK_D^\top)}{Z_1} V_D \\
&= (1 - \mu) \underbrace{\mathrm{SA}(q, K, V)}_{\text{attention}} + \mu \underbrace{\mathrm{SA}(q, K_D, V_D)}_{\text{shift vector}}
\end{aligned}
\tag{2}
$$

where $\mu(q, K_D, K) = Z_1/(Z_1 + Z_2)$, with $Z_1(q, K_D) = \sum_{i=1}^{T_D} \exp(qK_D^\top)_i$ and $Z_2(q, K) = \sum_{j=1}^{T} \exp(qK^\top)_j$. Equation 2 shows that the self-attention over the prompt context $C$ can be decomposed into two terms. For the former "standard attention", it is the self-attention over the query tokens, which is independent of the ICDs. While for the latter "shift vector", it is the shift effects caused by the ICDs to shift the query space into the answer space, and such effects are calculated as the attention between the ICDs and the query $q$. In in-context settings, demonstrations primarily affect where the model attends by injecting additional key-value evidence. Approximations that add a vector to the attention output behave like value-stream shifts (Peng et al., 2024; Jiang et al., 2025); however, they do not directly calibrate the attention distribution itself. To address this, we propose an intervention that directly calibrates the attention distribution via a learnable logit-level bias.

### 3.2. In-Context Attention Calibration

**Low-rank Logit-level Adapter.** At layer $l$, and attention head $h$, the standard attention logits are computed using the scaled dot product between queries $Q^{(l,h)}$ and keys $K^{(l,h)}$:

$$S^{(l,h)} = \frac{Q^{(l,h)}\left(K^{(l,h)}\right)^\top}{\sqrt{d_h}}, \qquad (3)$$

where $Q^{(l,h)}, K^{(l,h)} \in \mathbb{R}^{T \times d_h}$ and $S^{(l,h)} \in \mathbb{R}^{T \times T}$. Therefore, the head output is computed by:

$$O^{(l,h)} = \mathrm{softmax}\left(S^{(l,h)}\right) V^{(l,h)}. \qquad (4)$$

Unlike prior studies that approximate demonstration effects by adding a vector to the attention output, we intervene

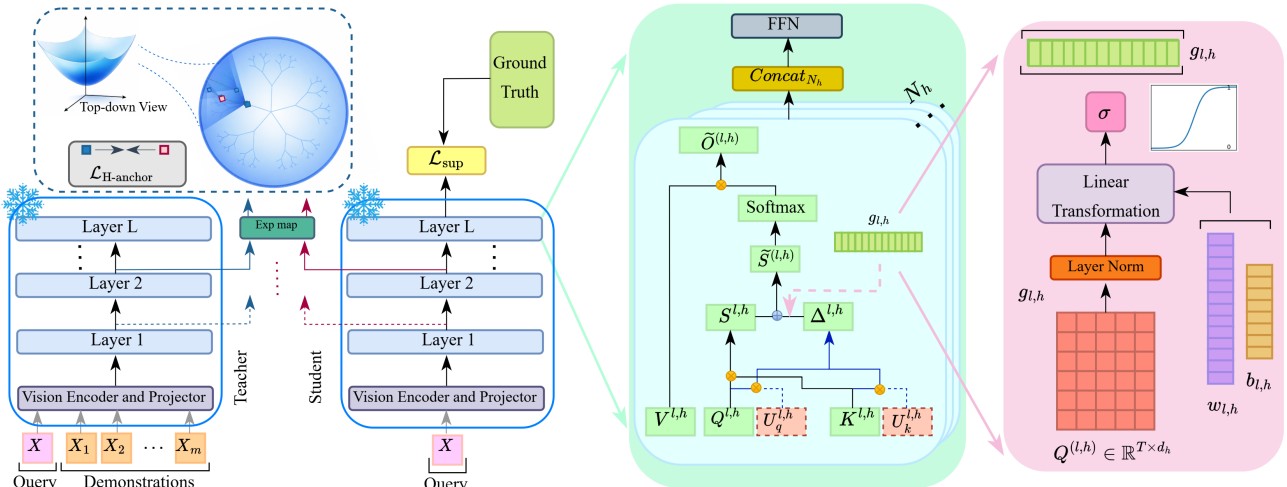

*Figure 1.* **Overview of Hyper-ICL.** A frozen teacher processes the full demonstration-conditioned prompt, while the student receives only the query at inference time. Hyper-ICL reconstructs demonstration effects by calibrating attention through a parameter-efficient low-rank logit-level adapter, whose strength is controlled token-wise by a query-adaptive gate $g_{l,h}$ across layers and heads. A layer-wise hyperbolic anchor distillation loss further aligns intermediate student features to the teacher in Lorentz space via geodesic distance, preserving the demonstration–query relationships for demonstration-free inference.

directly on the pre-softmax logits to steer the attention distribution. Concretely, we do this by adding a learnable bias at the logit level as below:

$$\widetilde{S}^{(l,h)} = S^{(l,h)} + \mathrm{Diag}(g_{l,h})\,\Delta^{(l,h)}, \qquad (5)$$

where $\Delta^{(l,h)} \in \mathbb{R}^{T \times T}$ is a learnable logit-bias matrix and $g_{l,h}$ is a query-adaptive token-wise modulation vector that controls the strength of the intervention for each query token. To keep this intervention parameter-efficient, we parameterize $\Delta^{(l,h)}$ with a compact low-rank bilinear form with rank $r \ll d_h$ as:

$$A^{(l,h)} = Q^{(l,h)} U_q^{(l,h)} \in \mathbb{R}^{T \times r},$$
$$B^{(l,h)} = K^{(l,h)} U_k^{(l,h)} \in \mathbb{R}^{T \times r}, \qquad (6)$$

where $U_q^{(l,h)}, U_k^{(l,h)} \in \mathbb{R}^{d_h \times r}$ are learnable projection matrices. Then, the logit bias is computed as:

$$\Delta^{(l,h)} = \frac{A^{(l,h)} B^{(l,h)T}}{\sqrt{r}} \in \mathbb{R}^{T \times T}. \qquad (7)$$

It should be noted that Equation 7 is equivalent to a low-rank bilinear term $Q(U_q^{(l,h)} U_k^{(l,h)\top}) K^\top$ but parameterized with only $2d_h r$ parameters per head while by considering $M^{(l,h)} \triangleq U_q^{(l,h)} U_k^{(l,h)\top}$, learning an unconstrained matrix $M^{(l,h)}$ requires specifying all of its entries, which is equal to $d_h \times d_h = d_h^2$ parameters. Finally, the output of head $h$ under the biased logits is computed as:

$$\widetilde{O}^{(l,h)} = \mathrm{softmax}\!\left(\widetilde{S}^{(l,h)}\right) V^{(l,h)}. \qquad (8)$$

This directly steers the attention distribution toward tokens that would have been emphasized by demonstrations.

**Query-adaptive Token-wise Modulation.** Our experiments show that the utility of logit intervention varies across the transformer depth and attention heads. Earlier layers primarily support perceptual grounding, whereas later layers contribute more to compositional reasoning and generation. Consequently, applying a uniform intervention strength across all layers can induce unnecessary interference, especially for inputs that do not require attention correction. To address this, we propose a token-wise modulation mechanism that adaptively scales the intervention strength for each layer and head based on the current query representation.

For layer $l$ and head $h$, let the query matrix be $Q^{(l,h)} \in \mathbb{R}^{T \times d_h}$. We compute a query-adaptive token-wise modulation vector $g_{l,h} \in (0,1)^T$ by applying a shared affine mapping to each query token after layer normalization:

$$g_{l,h} = \sigma\!\Big(\mathrm{LN}\big(Q^{(l,h)}\big)\,w_{l,h} + b_{l,h}\mathbf{1}\Big), \qquad (9)$$

where $w_{l,h} \in \mathbb{R}^{d_h}$ and $b_{l,h} \in \mathbb{R}$ are learnable parameters specific to layer $l$ and head $h$, $\mathbf{1} \in \mathbb{R}^T$ is the all-ones vector, $\mathrm{LN}(\cdot)$ denotes layer normalization, and $\sigma(\cdot)$ is the element-wise sigmoid function. This design produces one modulation coefficient per query token, enabling fine-grained and input-dependent control over the intervention magnitude within each attention head. Therefore, we not only calibrate *where* attention goes through logit modulations, but also make the calibration strength *query-adaptive*. This enables the model to selectively activate the intervention in the layers and heads where it is most helpful for the current input, while suppressing unnecessary interference elsewhere.

## 3.3. Layer-wise Hyperbolic Anchor Distillation

Building on the token-wise modulated logit-bias attention in Equation 5, we introduce an intermediate alignment regularizer that matches the student's internal representations to those of a frozen teacher model that observes full demonstrations. Concretely, the teacher processes the full prompt context $C = [X_D; X]$, while the student processes only the query tokens $X$ and relies on the learned logit intervention to recover the demonstration-conditioned behavior. This design provides a structured distillation signal beyond output-level matching, encouraging consistent layer-wise transformations. As the teacher encodes many demonstration-conditioned relations among query tokens, a Euclidean penalty can prioritize pointwise matching while distorting relative similarity ordering when many relations compete in a low-dimensional embedding. Negative curvature offers higher effective representational capacity and mitigates distortion in low-dimensional embeddings; thus, geodesic alignment can better preserve the multi-scale distances that reflect how demonstrations reshape intermediate features. This is especially relevant when the student must reconstruct those relations without seeing demonstrations.

**Lorentzian Representation of Layer Features.** Let $h_i^{(l)} \in \mathbb{R}^d$ denote the student hidden state of token index $i$ at layer $l \in \{1, \ldots, L\}$, and let $h_i'^{(l)} \in \mathbb{R}^d$ denote the corresponding teacher hidden state for the same query token, computed under the full context $C$. To capture hierarchical structure across layers in a geometry-aware manner, we align student and teacher representations in a hyperbolic space, whose distances naturally preserve layered, tree-like relationships. Specifically, we adopt the Lorentz model of hyperbolic space with constant negative curvature magnitude $\kappa > 0$. The Lorentzian inner product on $\mathbb{R}^{d+1}$ is defined as:

$$\langle p, q \rangle_L \triangleq -p_0 q_0 + \sum_{t=1}^{d} p_t q_t, \tag{10}$$

and the corresponding $d$-dimensional hyperboloid manifold is expressed as:

$$\mathbb{H}_\kappa^d \triangleq \left\{ p \in \mathbb{R}^{d+1} : \langle p, p \rangle_L = -\frac{1}{\kappa}, \ p_0 > 0 \right\}. \tag{11}$$

We use the canonical base point $o \triangleq \left( \sqrt{1/\kappa}, 0, \ldots, 0 \right)^\top$ as the origin of the manifold. Next, we map Euclidean hidden states onto $\mathbb{H}_\kappa^d$ using the exponential map at $o$. To ensure scale stability and consistency with the normalization used in Equation 9, we first apply layer normalization to each hidden state as follows:

$$u_i^{(l)} \triangleq \text{LN}\left( h_i^{(l)} \right) \in \mathbb{R}^d, \qquad \bar{u}_i^{(l)} \triangleq \left( 0, u_i^{(l)} \right)^\top \in \mathbb{R}^{d+1}. \tag{12}$$

Since the tangent space at $o$ can be identified with vectors whose first coordinate is zero, the padded vector $\bar{u}_i^{(l)}$ is a valid tangent direction at $o$. We then apply the Lorentz exponential map as:

$$P_i^{(l)} \triangleq \exp_o^\kappa(u_i^{(l)})$$

$$= \cosh(\sqrt{\kappa}\|u_i^{(l)}\|)o + \frac{\sinh(\sqrt{\kappa}\|u_i^{(l)}\|)}{\sqrt{\kappa}\|u_i^{(l)}\|} \bar{u}_i^{(l)} \in \mathbb{H}_\kappa^d, \tag{13}$$

where $\| \cdot \|$ denotes the Euclidean norm in $\mathbb{R}^d$, and the fraction is defined as 1 when $\|u_i^{(l)}\| = 0$. Similarly, for the teacher, we define:

$$P_i'^{(l)} \triangleq \exp_o^\kappa\left( u_i'^{(l)} \right) \in \mathbb{H}_\kappa^d, \tag{14}$$

where $u_i'^{(l)} \triangleq \text{LN}(h_i'^{(l)}) \in \mathbb{R}^d$ and $\bar{u}_i'^{(l)} \triangleq (0, u_i'^{(l)})^\top \in \mathbb{R}^{d+1}$.

**Hyperbolic-Based Regularization.** Given the hyperbolic embeddings $P_i^{(l)}$ and $P_i'^{(l)}$, we measure alignment using the Lorentz geodesic distance on $\mathbb{H}_\kappa^d$:

$$d_L(P', P) = \sqrt{\frac{1}{\kappa}} \, \text{arcosh}\left( -\kappa \left\langle P', P \right\rangle_L \right), \tag{15}$$

where the argument $-\kappa \left\langle P', P \right\rangle_L$ is guaranteed to be greater than or equal to 1 for valid points on the hyperboloid, making the distance well-defined. We then define a layer-wise hyperbolic anchor distillation loss that aligns student and teacher representations across all transformer layers:

$$\mathcal{L}_{\text{H-anchor}} = \frac{1}{L} \sum_{l=1}^{L} \frac{1}{T} \sum_{i=1}^{T} d_L^2\left( P_i'^{(l)}, P_i^{(l)} \right). \tag{16}$$

This loss provides explicit intermediate supervision, encouraging the student to reproduce the teacher's demonstration-conditioned layer-wise transformations, rather than only matching final outputs. Intuitively, since the teacher observes $X_D$ while the student does not, minimizing Equation 16 forces the student to recover teacher-like representations using only the learned attention-logit interventions.

In addition, we employ a standard cross-entropy loss function, $\mathcal{L}_{\text{sup}}$, as the main supervised loss objective to enhance student model's predictions on downstream tasks. Therefore, the final training objective combines these losses as a weighted sum:

$$\mathcal{L} = \mathcal{L}_{\text{H-anchor}} + \lambda \mathcal{L}_{\text{sup}}, \tag{17}$$

where $\lambda \geq 0$ is a hyperparameter that balances task supervision and intermediate alignment. This overall objective ensures that Hyper-ICL learns to apply efficient logit-level calibration while preserving layer-wise behaviors consistent with query-conditioned inference.

## 4. Experiments

### 4.1. Implementation Details

We evaluate Hyper-ICL on two large-scale MLLMs, Idefics-9b (Laurençon et al., 2023) and Idefics2-8B-base (Laurençon et al., 2024). Idefics-9b adopts a cross-attention architecture, whereas Idefics2-8B-base follows a fully autoregressive design, representing two widely used architectural paradigms for vision-language modeling. Our experiments span a diverse set of benchmarks, including VQAv2

*Table 1.* Results of VQAv2, OK-VQA, and COCO on Idefics-9b and Idefics2-8B-base. **Bold numbers** represent the best results. In # Params (M), the first value denotes the number of trainable parameters (in millions), and the value in parentheses reports the relative factor compared to Hyper-ICL.

| Model | Type | Method | # Params (M) | VQAv2 | OK-VQA | COCO |
|---|---|---|---|---|---|---|
| **Idefics-9b** | Direct ICDs | Zero-shot | - | 29.25 | 30.54 | 63.06 |
| | | 32-shot ICL | - | 56.18 | 48.48 | 105.89 |
| | | RICES | - | 58.07 | 51.11 | 110.64 |
| | Non-Learnable | FV | - | 30.21 | 31.02 | 74.01 |
| | | TV | - | 43.68 | 32.68 | 84.72 |
| | Learnable | LoRA | 25.0 (×21.2) | 55.60 | 47.06 | 97.75 |
| | | LIVE | 0.13 (×0.11) | 53.71 | 46.05 | 112.76 |
| | | MimIC | 0.26 (×0.22) | 59.64 | 52.05 | 114.89 |
| | | Hyper-ICL | 1.18 (×1) | **62.08** | **55.31** | **117.44** |
| **Idefics2-8b** | Direct ICDs | Zero-shot | - | 55.39 | 43.08 | 40.00 |
| | | 8-shot ICL | - | 66.20 | 57.68 | 122.51 |
| | | RICES | - | 66.44 | 55.73 | 111.44 |
| | Non-Learnable | FV | - | 36.47 | 34.58 | 75.24 |
| | | TV | - | 47.12 | 38.27 | 87.61 |
| | Learnable | LoRA | 17.6 (×14.9) | 66.54 | 55.05 | 116.69 |
| | | LIVE | 0.13 (×0.11) | 67.60 | 54.86 | 126.04 |
| | | MimIC | 0.26 (×0.22) | 69.29 | 58.74 | 132.87 |
| | | Hyper-ICL | 1.18 (×1) | **71.17** | **62.24** | **135.66** |

*Table 2.* Results evaluated on more challenging tasks.

| Model | Method | Flickr30k | MME | SEED |
|---|---|---|---|---|
| Idefics-9b | Zero-shot | 49.17 | 55.36 | 27.56 |
| | ICL | 63.41 | 52.11 | 28.30 |
| | Hyper-ICL | **75.96** | **65.46** | **31.87** |
| Idefics2-8B-base | Zero-shot | 53.04 | 74.80 | 12.91 |
| | ICL | 84.57 | 71.10 | 47.90 |
| | Hyper-ICL | **93.79** | **82.13** | **48.31** |

2024), we use 10,000 validation examples from VQAv2 and the full validation splits for OK-VQA and COCO. We employ the AdamW optimizer with a learning rate of $5 \times 10^{-3}$, coupled with a cosine annealing scheduler with warmup, allocating 10% of the total steps for warmup. All results are reported from the best-performing epoch.

**4.2. Comparison with Existing Methods**

We compare Hyper-ICL with the following methods:

**Direct use of ICDs.** It is evaluated under three settings: zero-shot, few-shot, and Retrieval-based In-Context Examples Selection (RICES) (Yang et al., 2022). For few-shot ICL, we use 32/8-shot for Idefics-9B and Idefics2-8B-base, respectively. We also compare Hyper-ICL with RICES, which retrieves visually similar support images for each query by matching features extracted from a frozen pre-trained encoder.

**Non-learnable vector-based ICDs.** Task Vector (TV) (Brown et al., 2020) and Function Vector (FV) (Todd et al., 2023) that extract an in-context vector from examples and inject it into the model's hidden states during inference.

**Learnable use of ICDs.** We compare Hyper-ICL with LoRA (Hu et al., 2022), LIVE (Peng et al., 2024), and MimIC (Jiang et al., 2025), where LIVE and MimIC are trainable ICV methods that distill few-shot ICD effects into learnable shift vectors under the same few-shot setting. For LoRA, we follow the standard setup and apply it to all attention layers in both the vision and language encoders.

Table 1 compares Hyper-ICL against a broad set of baselines on two MLLMs and three datasets. Overall, prior non-trainable ICD selection methods remain consistently below the standard 32-shot ICL setting. For instance, while RICES improves over random selection on all datasets for Idefics-9B, it becomes less reliable on Idefics2-8B-base, where its performance on OK-VQA and COCO falls below random selection. This behavior stems from architectural differences: nearest-neighbor retrieval aligns well with Idefics-9B's cross-attention fusion, whereas Idefics2-8B-base's fully autoregressive decoding benefits more from context diversity than visual similarity. In contrast to non-trainable approaches, trainable methods consistently improve performance across both backbones, often approaching the effectiveness of few-shot ICL. Notably, Hyper-ICL achieves the best results among all learnable baselines on both MLLMs,

(Goyal et al., 2017), OK-VQA (Marino et al., 2019), COCO Caption (Chen et al., 2015), Flickr30k (Young et al., 2014), MME (Fu et al., 2025), and SEED-Bench (Li et al., 2024a). To compare Hyper-ICL against prior SOTA methods, we focus on three widely used datasets: VQAv2, OK-VQA, and COCO Caption. VQAv2 targets open-ended visual question answering and contains 4,437,570 question-answer pairs in the training split, along with 2,143,540 pairs in the validation split. OK-VQA is designed to evaluate models that require external knowledge, comprising 14,055 question-answer pairs, with 9,009 for training and 5,046 for validation. COCO Caption is a standard benchmark for image captioning built on the MS COCO image collection, providing multiple human-annotated descriptions per image to capture diverse yet semantically consistent views of the visual content. We further extend our evaluation to Flickr30k (Young et al., 2014), which includes 31,000 images each paired with five captions, MME (Fu et al., 2025), a comprehensive benchmark with 14 subtasks assessing perceptual and cognitive capabilities, and SEED-Bench (Li et al., 2024a), a multi-modal benchmark comprising 24,000 multiple-choice questions.

For the training stage, we randomly sample 1,000 instances from each dataset to form the training set. In addition, we randomly select 32 samples as ICDs for Idefics-9b and 8 samples for Idefics2-8B-base, along with one separate sample used as the query input. Following prior evaluation protocols (Li et al., 2024b; Liu et al., 2023; Peng et al.,

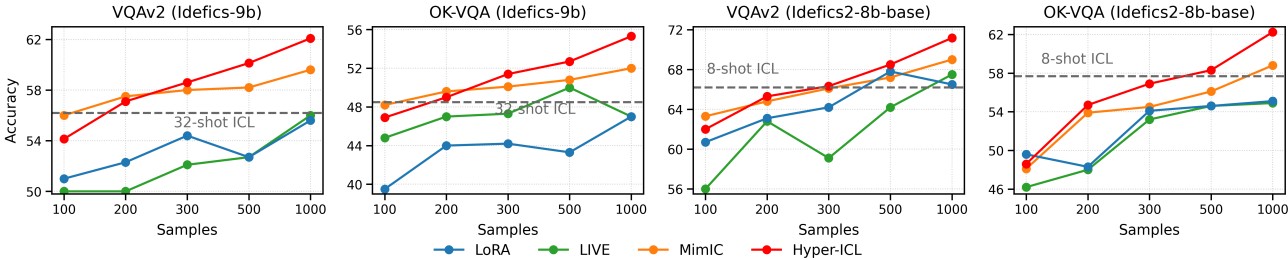

*Figure 2.* Comparison of Hyper-ICL with LoRA, LIVE, and MimIC across training sample sizes on VQAv2 and OK-VQA using Idefics-9B and Idefics2-8B-base. Dashed horizontal lines denote standard ICL baselines, 32-shot for Idefics-9B and 8-shot for Idefics2-8B-base.

highlighting its robustness and stability across diverse architectures and evaluation benchmarks. We further evaluate Hyper-ICL on additional MLLM backbones in Appendix A.

### 4.3. Generalize to More Challenging Benchmarks

We further evaluate Hyper-ICL on more diverse and challenging tasks, including Flickr30k (Young et al., 2014), MME (Fu et al., 2025), and SEED-Bench (Li et al., 2024a). Table 2 shows that Hyper-ICL consistently improves over zero-shot and standard ICL across both backbones, demonstrating that our logit-level intervention and layer-wise distillation generalize well across diverse multimodal tasks and evaluation suites. Due to computational constraints, we run 2-shot ICL on MME and SEED-Bench for Idefics2-8B-base. While Hyper-ICL remains lightweight and does not require storing long KV caches (in contrast to few-shot ICL), we also report Hyper-ICL results with 2 shots on MME and SEED-Bench to match the ICL setting for a fair comparison. To further assess transferability, Appendix B evaluates Hyper-ICL on held-out benchmarks within the same task family.

### 4.4. Ablation Studies and Discussions

**Size of Training Set.** Figure 2 analyzes how performance scales with the number of training samples for three trainable SOTA ICL methods, along with our proposed Hyper-ICL, on VQAv2 and OK-VQA using two MLLM backbones. As the training set grows, all methods improve, confirming that supervised adaptation is beneficial in the low-data regime. However, Hyper-ICL consistently achieves higher accuracy than LoRA, LIVE, and MimIC across most sample budgets and settings. Notably, Hyper-ICL surpasses the standard few-shot ICL baseline with substantially fewer training examples. The advantage is especially clear on OK-VQA, where external knowledge demands capturing demonstration–query relationships and hierarchical dependencies, and Hyper-ICL delivers a clear margin over prior trainable methods.

**Number of Demonstrations.** To further analyze shot sensitivity, we extend the Idefics-9B study beyond the 32-shot setting to 4, 8, 16, 48, and 64 shots. Table 3 shows that both standard ICL and Hyper-ICL improve as the number

*Table 3.* Effect of the number of demonstrations on Idefics-9B.

| Bench. | Method | 4 | 8 | 16 | 32 | 48 | 64 |
|---|---|---|---|---|---|---|---|
| VQAv2 | ICL | 53.72 | 54.24 | 55.70 | 56.18 | 56.67 | 56.71 |
| | Hyper-ICL | **57.87** | **58.81** | **59.73** | **62.08** | **62.61** | **62.90** |
| OK-VQA | ICL | 46.11 | 46.79 | 47.70 | 48.48 | 48.68 | 48.60 |
| | Hyper-ICL | **50.21** | **51.79** | **53.22** | **55.31** | **55.53** | **55.26** |

of demonstrations increases, but the gains become marginal beyond 32 shots and, on OK-VQA, slightly decline at 64 shots. This reveals a clear saturation effect: once the teacher context is sufficiently informative, additional demonstrations provide limited new signal and can introduce more variability. Importantly, Hyper-ICL remains clearly superior to standard ICL across all shot counts.

**Effect of Training Losses.** To assess the contribution of task supervision and intermediate distillation, we ablate the training objectives on Idefics-9B in Table 4. Supervised adaptation with $\mathcal{L}_{\text{sup}}$ substantially improves over zero-shot inference, confirming the effectiveness of lightweight task-specific training. Adding intermediate alignment brings further gains, with the proposed hyperbolic anchor distillation loss $\mathcal{L}_{\text{H-anchor}}$ outperforming Euclidean alignment $\mathcal{L}_{\text{L}_2\text{-anchor}}$ and achieving the best results on both VQAv2 and OK-VQA. This supports our motivation that negative curvature provides a more suitable geometry for preserving dense demonstration-conditioned relations with lower distortion in a low-dimensional space. Importantly, $\mathcal{L}_{\text{H-anchor}}$ alone also outperforms $\mathcal{L}_{\text{sup}}$ alone, indicating that the improvement is not merely due to ordinary supervised task adaptation, but also comes from recovering ICL-induced intermediate structure from the teacher. The best performance is obtained when both objectives are combined, suggesting that hyperbolic alignment transfers demonstration-conditioned relational geometry, while task supervision further improves final prediction accuracy. We further provide a direct hyperbolicity analysis in Appendix C, showing that the hyperbolic anchor better preserves the teacher's demonstration-conditioned hidden-state geometry than the Euclidean anchor.

**Analysis of Adapter Rank and Query-Adaptive Token-**

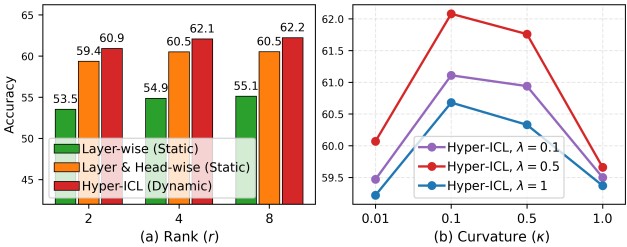

*Figure 3.* Ablation of Hyper-ICL hyperparameters and architectural choices on Idefics-9B (VQAv2). (a) Effect of the low-rank adapter rank $r$ for logit-level attention calibration, comparing static interventions (layer-wise, layer & head-wise) with our query-adaptive token-wise modulated Hyper-ICL. (b) Sensitivity of the hyperbolic anchor distillation to curvature $\kappa$ and the supervision weight $\lambda$, showing the best performance at $\kappa = 0.1$ and $\lambda = 0.5$.

**wise Modulation.** Figure 3(a) analyzes the effect of the adapter rank ($r$) in our logit-level module and compares static attention interventions against the proposed token-wise modulation. Increasing $r$ improves performance for all variants; however, beyond $r > 4$, the gains become marginal while incurring additional computational cost. Also, the rank sensitivity is evaluated under multiple intervention strategies, ensuring a fair comparison across methods. Hyper-ICL consistently outperforms both static baselines (layer-wise and layer & head-wise interventions), reaching 60.9, 62.1, and 62.2 for $r = 2, 4, 8$, respectively. These results highlight the advantage of token-wise intervention modulation $g_{l,h}$, which enables query-adaptive calibration while avoiding uniform, query-agnostic modulations.

We further analyze the learned query-adaptive gates in Appendix D, showing that the modulation remains non-trivial across layers and correlates positively with the degree of demonstration-induced attention redistribution. In addition, Appendix E compares our logit-level attention calibration with an output-level correction, confirming that directly reshaping attention logits is more effective than correcting representations after attention aggregation.

**Analysis of Curvature and Loss-Weight Sensitivity.** Figure 3(b) studies the hyperparameter sensitivity of the proposed hyperbolic anchor distillation by sweeping the curvature $\kappa$ and the supervision weight ($\lambda$) on VQAv2 with Idefics-9B. For all $\lambda$, accuracy is maximized at an intermediate curvature, with a clear optimum at $\kappa = 0.1$. When $\kappa$ is too small (0.01), the geometry becomes close to Euclidean, weakening the hierarchy-aware layer alignment and reducing performance. Conversely, a large curvature ($\kappa = 1.0$) makes geodesic distances overly steep, which can destabilize optimization and degrade accuracy across all $\lambda$. Overall, the best trade-off is achieved with $\kappa = 0.1$ and $\lambda = 0.5$, indicating that effective distillation requires both a well-conditioned hyperbolic geometry and a balanced coupling between intermediate alignment and task supervision.

*Table 4.* Ablation of Hyper-ICL training objectives on Idefics-9B for VQAv2 and OK-VQA.

| Task | Zero-shot | $\mathcal{L}_{sup}$ | $\mathcal{L}_{sup} + \mathcal{L}_{L_2\text{-anchor}}$ | $\mathcal{L}_{H\text{-anchor}}$ | $\mathcal{L}_{sup} + \mathcal{L}_{H\text{-anchor}}$ |
|---|---|---|---|---|---|
| **VQAv2** | 29.25 | 57.11 | 59.64 | 60.53 | **62.08** |
| **OK-VQA** | 30.54 | 49.23 | 53.76 | 53.78 | **55.31** |

*Table 5.* Comparison of caption hallucination across methods on COCO. Lower CHAIR scores reflect fewer hallucinations, and higher Recall reflects stronger coverage and better performance.

| Metric | Z-Shot | 32-ICL | TV | FV | LoRA | LIVE | MimIC | Hyper-ICL |
|---|---|---|---|---|---|---|---|---|
| **CHAIRs** | 5.93 | 16.78 | 8.88 | 28.26 | 17.42 | 8.65 | 8.51 | 8.29 |
| **CHAIRi** | 5.58 | 9.77 | 7.50 | 25.44 | 11.55 | 6.05 | 5.74 | 5.63 |
| **Recall** | 30.72 | 42.59 | 36.22 | 27.69 | 42.93 | 42.84 | 43.30 | **44.04** |

**Hallucination Analysis.** Table 5 presents caption hallucination on COCO using CHAIRs and CHAIRi (Rohrbach et al., 2018), which quantify object hallucination at the sentence level and the instance level, respectively. CHAIRs measures the percentage of captions that contain at least one object not grounded in the image, while CHAIRi measures the percentage of hallucinated object mentions among all object mentions. We additionally report Recall, which reflects how many ground-truth objects are correctly covered by the generated caption and thus captures descriptiveness. Results show that Hyper-ICL yields fewer hallucinations than all non-zero-shot baselines while achieving the highest Recall. Compared to zero-shot, Hyper-ICL shows a slight increase in hallucinations, consistent with the limitations of ICL where using more shots can amplify hallucinations (Qin et al., 2024; Shukor et al., 2024); however, Hyper-ICL achieves substantially higher Recall than the other approaches. The 32-shot ICL results further support this trend, as hallucinations rise markedly (CHAIRs 16.78, CHAIRi 9.77), aligning with stronger but less controlled demonstration-induced shifts. Figure 4 further provides qualitative evidence that Hyper-ICL reduces hallucinations by keeping generation anchored to visual evidence, compared with standard ICL, LoRA, and LIVE. In the VQA examples, LIVE, LoRA, and standard ICL either output incorrect textual content or latch onto superficial cues, such as misreading the sign text, predicting the wrong purse color, or selecting an incorrect street as straight ahead. Overall, this qualitative behavior is consistent with Table 5, suggesting that our logit-level attention calibration and query-adaptive token-wise modulation selectively correct attention when needed.

**Inference Efficiency Analysis.** Table 6 quantifies the inference efficiency of Hyper-ICL by comparing its FLOPs and runtime against zero-shot and standard few-shot ICL. Hyper-ICL remains close to zero-shot, requiring 0.955T FLOPs and 59.32 ms, while ICL incurs a sharp cost increase as the number of demonstrations grows due to long input context and heavy attention computation. These results confirm that Hyper-ICL preserves demonstration-level

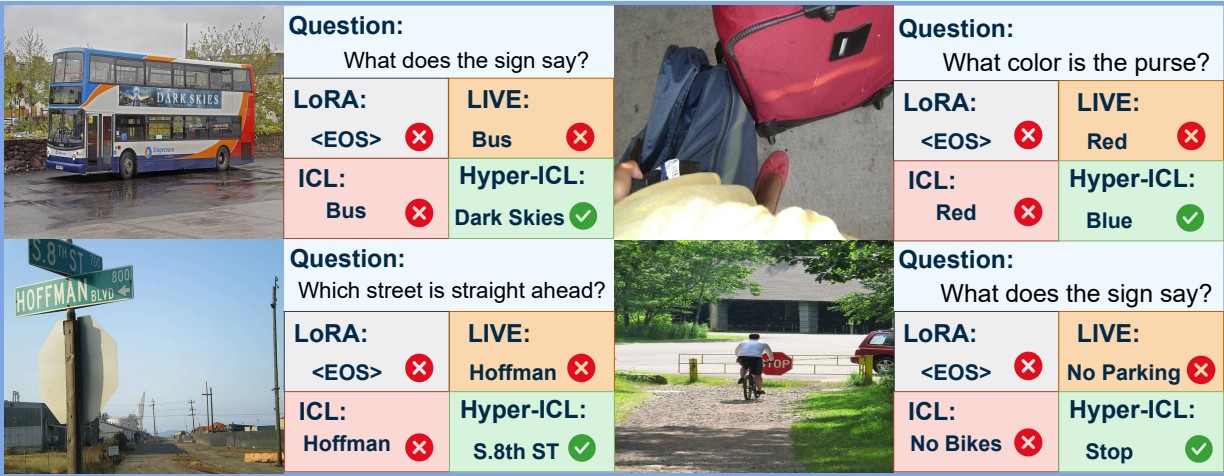

*Figure 4.* Qualitative comparison of hallucination behavior across methods on VQA. Hyper-ICL stays visually grounded on representative queries (e.g., sign text, purse color, and road direction), while standard ICL, LoRA, and LIVE produce ungrounded content.

*Table 6.* Analysis of inference FLOPs and runtime.

| Metric | Hyper-ICL | 0-Shot | 8-shot | 16-shot | 32-shot |
|---|---|---|---|---|---|
| FLOPs (T) | 0.955 | 0.935 | 6.375 | 12.341 | 23.364 |
| Runtime (ms) | 59.32 | 56.69 | 158.21 | 266.13 | 468.55 |

benefits while enabling demonstration-free inference with near zero-shot efficiency.

## 5. Conclusion

In this work, we present Hyper-ICL, a training-based framework for demonstration-free multimodal ICL that reconstructs ICD effects inside self-attention. Hyper-ICL introduces a parameter-efficient low-rank logit-level adapter that injects head-wise biases into attention logits, together with a query-adaptive token-wise modulation mechanism to enable query-adaptive calibration across layers and heads. To transfer demonstration-conditioned representations, we propose a layer-wise hyperbolic anchor distillation loss, aligning student representations to a frozen demonstration-conditioned teacher via geodesic distance. By removing ICDs at inference, it shortens context length and reduces latency while retaining the benefits of few-shot prompting. Experiments on Idefics-9B and Idefics2-8B over VQAv2, OK-VQA, and COCO Caption show consistent gains and improved stability over direct ICD prompting, vector-based ICV baselines, and trainable SOTA methods.

## Acknowledgment

This material is based upon work supported by the National Science Foundation under Grant Numbers CNS-2232048, and CNS-2204445.

## Impact Statement

This work aims to improve the efficiency and practicality of multimodal ICL by eliminating the need for demonstration tokens at inference time. By reconstructing demonstration effects through a lightweight adapter, Hyper-ICL achieves near zero-shot inference cost while retaining the benefits of few-shot prompting, substantially reducing latency and computational overhead. This efficiency can support greener AI deployment by lowering inference-time energy consumption, which is especially important for large-scale vision-language systems. The low-latency profile of Hyper-ICL may also broaden the use of multimodal models in real-time applications such as assistive technologies, medical decision support, and interactive vision-language systems, where long demonstration-based prompts are often impractical.

A limitation of the proposed framework is that the adapter is learned for a given task or domain setting. Therefore, switching to substantially different tasks, domains, or model backbones may require retraining or re-calibrating the adapter. As with other multimodal learning systems, deployment in high-stakes settings should include careful validation, robustness testing, and human oversight.

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

## A. Generalize to Additional MLLMs

To further evaluate the robustness and scalability of Hyper-ICL across different multimodal architectures, we additionally conduct experiments on two recent MLLMs: LLaVA-Interleave-7B (Li et al., 2025b) and LLaVA-OneVision (Qwen2-7B) (Li et al., 2025a). Table 7 shows that Hyper-ICL consistently outperforms zero-shot, standard few-shot ICL, and the recent MimIC method (Jiang et al., 2025) on both VQAv2 (Goyal et al., 2017) and OK-VQA (Marino et al., 2019). These results further demonstrate that the proposed logit-level attention calibration and hyperbolic distillation generalize effectively across diverse MLLM architectures.

*Table 7.* Results evaluated on additional MLLMs.

| Backbone | Method | VQAv2 | OK-VQA |
|---|---|---|---|
| LLaVA-Interleave-7B | Zero-shot | 13.02 | 5.10 |
| | 8-shot ICL | 68.19 | 43.84 |
| | MimIC | 74.40 | 52.29 |
| | Hyper-ICL | **76.51** | **56.77** |
| LLaVA-OneVision (Qwen2-7B) | Zero-shot | 71.75 | 48.19 |
| | 8-shot ICL | 78.70 | 66.59 |
| | MimIC | 81.22 | 69.43 |
| | Hyper-ICL | **82.24** | **75.32** |

## B. Held-out Benchmark Transfer

We further evaluate whether Hyper-ICL can transfer to unseen benchmarks within the same task family. We conduct two held-out transfer experiments on Idefics2-8B-base (Laurençon et al., 2024). In the first setting, the model is trained on VQAv2 and evaluated on the unseen OK-VQA benchmark. In the second setting, the model is trained on COCO Caption (Chen et al., 2015) and evaluated on Flickr30k (Young et al., 2014). For a fair comparison, the 8-shot ICL baseline also uses demonstrations drawn from the source dataset rather than the target dataset. As shown in Table 8, Hyper-ICL substantially improves over both zero-shot inference and standard 8-shot ICL on unseen benchmarks within the same task family. These results indicate that the learned attention calibration is not purely dataset-specific and can capture transferable demonstration-conditioned behavior across related benchmarks.

*Table 8.* Held-out benchmark transfer results on Idefics2-8B-base.

| Train | Test | Zero-shot | 8-shot ICL | Hyper-ICL |
|---|---|---|---|---|
| COCO | Flickr30k | 53.04 | 73.26 | **87.42** |
| VQAv2 | OK-VQA | 43.08 | 51.43 | **59.11** |

## C. Additional Analysis of Hyperbolic Structure

In this section, we provide a direct empirical analysis of the hyperbolic anchor loss to support our assumption that multimodal ICL induces dense demonstration-conditioned relations among query tokens. Specifically, we examine this geometric assumption by measuring whether the intermediate hidden-state representations exhibit a non-uniform, tree-like relational structure.

We quantify this structure using relative $\delta$-hyperbolicity, which measures how closely a metric space satisfies the Gromov four-point condition. Smaller values indicate that pairwise metric relations are closer to those of a tree-like geometry, while larger values indicate a flatter structure. Following (Khrulkov et al., 2020), we use the scale-invariant score as follows:

$$\delta_{\text{rel}}(X) = \frac{2\delta(X)}{\text{diam}(X)} \in [0, 1], \tag{18}$$

where $\delta(X)$ is the estimated Gromov hyperbolicity of the metric space $X$.

We conduct this analysis on Idefics2-8B-base using 1,000 VQAv2 samples. For each sample, we construct the metric space from query-token hidden states in the four middle transformer layers. We compare three settings: the demonstration-conditioned teacher hidden states, the student hidden states trained with the Euclidean anchor loss, and the student hidden states trained with the proposed hyperbolic anchor loss. For a fair comparison, we use the same sampled four-point tuples across all settings when estimating $\delta(X)$. Specifically, we estimate $\delta(X)$ using the efficient four-point procedure adopted

in (Khrulkov et al., 2020; Fournier et al., 2015): we sample 1,000 four-tuples, compute the four-point $\delta$ value for each, take the maximum sampled value, normalize by the diameter, and report mean $\pm$ std.

*Table 9.* Relative $\delta$-hyperbolicity of intermediate hidden-state representations on Idefics2-8B-base. Lower values indicate a metric geometry closer to a tree-like structure.

| Setting | $\delta_{\mathbf{rel}} \downarrow$ |
|---|---|
| Teacher hidden states | $0.228 \pm 0.014$ |
| Student hidden states + Euclidean anchor | $0.314 \pm 0.023$ |
| Student hidden states + Hyperbolic anchor | $0.247 \pm 0.016$ |

As shown in Table 9, the demonstration-conditioned teacher hidden states exhibit a measurable degree of hyperbolic structure, suggesting that the teacher representations are not merely an unstructured set of hidden vectors. Instead, they encode a non-uniform relational geometry induced by multimodal demonstrations. The student trained with the proposed hyperbolic anchor loss achieves a substantially lower $\delta_{\mathrm{rel}}$ than the Euclidean-anchor student and more closely matches the teacher geometry. These results provide direct evidence that the gain from the hyperbolic anchor is not only due to adding an intermediate regularizer, but is also consistent with better preserving the demonstration-conditioned relational structure that Hyper-ICL is designed to reconstruct.

## D. Interpretability Analysis of Query-Adaptive Modulation

In Section 3.2, Hyper-ICL introduces a query-adaptive token-wise modulation vector $g_{l,h} \in (0,1)^T$ to control the strength of the logit-level intervention for each query token across layers and heads. In this section, we provide an additional analysis of the learned modulation behavior to examine whether $g_{l,h}$ provides meaningful, input-dependent control rather than collapsing to a nearly constant scalar.

For layer $l$, head $h$, and query token $t$, we denote the corresponding modulation value by $g_{l,h,t}$. Larger values indicate stronger activation of the learned logit-level bias for that token. We analyze the learned gates on Idefics-9B (Laurençon et al., 2023) using subsets of VQAv2 and OK-VQA. First, we compute the mean gate value in early layers (0-7) and late layers (24-31), together with the average per-layer standard deviation over heads and query tokens. This evaluates whether the gate saturates near 0 or 1, or instead preserves meaningful variation across the model.

Second, we examine whether the learned gate becomes larger when demonstrations induce stronger changes in attention. For each query, we run the model twice: once with the full demonstration-conditioned prompt and once with the query-only prompt. We define the demonstration-induced attention shift as:

$$\Delta A_{l,h,t} = \left\| A_{l,h,t}^{\mathrm{ICD}} - A_{l,h,t}^{\mathrm{noICD}} \right\|_1, \tag{19}$$

where $A_{l,h,t}^{\mathrm{ICD}}$ and $A_{l,h,t}^{\mathrm{noICD}}$ denote the attention distributions for token $t$ under the demonstration-conditioned and query-only settings, respectively. We then compute the Pearson correlation between $g_{l,h,t}$ and $\Delta A_{l,h,t}$ over all evaluated examples. A positive correlation indicates that the gate assigns stronger intervention weights to tokens, heads, and layers where demonstrations cause larger attention redistribution.

*Table 10.* Quantitative analysis of query-adaptive token-wise modulation on Idefics-9B. Reported metrics include the mean gate value in early and late layers, the average per-layer standard deviation, and the Pearson correlation between the learned gate $g$ and the demonstration-induced attention shift $\Delta A$.

| Dataset | Early-layer mean $g$ | Late-layer mean $g$ | Avg. per-layer std $g$ | Corr$(g, \Delta A)$ |
|---|---|---|---|---|
| VQAv2 | 0.38 | 0.46 | 0.17 | 0.41 |
| OK-VQA | 0.43 | 0.62 | 0.23 | 0.56 |

Table 10 supports the intended behavior of the modulation mechanism in three ways. First, the mean gate values do not collapse toward 0 or 1, and the non-trivial per-layer standard deviations show that the gate preserves meaningful variation rather than acting as a nearly constant scalar. Second, the gate values increase in later layers, which is consistent with the design motivation that deeper layers contribute more strongly to compositional reasoning and answer generation. This pattern is more pronounced on OK-VQA, where the late-layer mean gate reaches 0.62, reflecting the stronger reasoning and knowledge demands of this benchmark. Third, the positive correlations between $g$ and $\Delta A$ on both datasets provide direct evidence that the gate is activated more strongly when demonstrations induce larger attention redistribution.

# E. Output-level Offset vs. Logit-level Attention Calibration

We further analyze whether the benefit of Hyper-ICL can be recovered by applying a learnable correction after attention, rather than intervening on the attention logits. Although modifying the attention logits ultimately changes the attention output, the two operations are not equivalent. An output-level offset directly perturbs the resulting vector representation after the attention-weighted summation, whereas our logit-level intervention changes the attention distribution before the value aggregation. Therefore, Hyper-ICL explicitly controls which tokens are attended to and how strongly they compete under the softmax normalization.

This distinction is important in multimodal ICL, where demonstrations mainly affect the query by redistributing attention over relevant visual and textual tokens. Existing output-level methods, such as LIVE (Peng et al., 2024), inject learned layer-wise shift vectors into hidden activations, but they do not directly calibrate the attention probabilities. As shown in Table 1, Hyper-ICL consistently outperforms LIVE across VQAv2, OK-VQA, and COCO, indicating that output-level shifts are less effective at reconstructing demonstration-induced behavior.

To further analyze the effect of intervention placement, we additionally implement a variant that uses the same low-rank, query-adaptive module as Hyper-ICL but applies it to the post-attention output $O$ rather than to the pre-softmax logits $S$. As shown in Table 11, the attention-logit intervention used by Hyper-ICL leads to stronger results than applying the same module at the output level. This confirms that the improvement does not simply come from adding a learnable correction, but from applying the correction at the attention-logit level, where the model can directly reshape token competition and attention redistribution.

*Table 11.* Comparison between applying the same low-rank, query-adaptive module to the post-attention output and applying it at the attention-logit level in Hyper-ICL.

| Method | VQAv2 | OK-VQA |
|---|---|---|
| Intervention on $O$ | 58.42 | 50.18 |
| Hyper-ICL | **62.08** | **55.31** |

