# OpenReview forum: "Hyper-ICL: Attention Calibration with Hyperbolic Anchor Distillation for Multimodal In-Context Learning"
_ICML.cc/2026/Conference — ICML 2026 regular_

### Official Review · Reviewer_S8Jk · 2026-03-12

**Soundness:** 2
**Presentation:** 2
**Significance:** 3
**Originality:** 3
**Overall Recommendation:** 4
**Confidence:** 3

**Summary:**

This paper introduces Hyper-ICL, a novel training-based framework designed to perform multi-modal In-Context Learning (ICL) without requiring any explicit context examples during inference. Traditional multi-modal ICL, while flexible, suffers from high latency due to lengthy interleaved image-text prompts and is unstable, as it is overly sensitive to the format and order of the provided examples. Hyper-ICL addresses these issues by reconstructing the effect of these examples directly within the model's internal mechanisms. It achieves this through three key technical contributions. Extensive experiments on six diverse multi-modal benchmarks demonstrate that Hyper-ICL consistently outperforms both vanilla ICL and existing state-of-the-art methods in terms of accuracy and stability, offering an efficient and robust alternative for deploying multi-modal large language models.

**Compliance With Llm Reviewing Policy:**

Affirmed.

**Final Justification:**

Soundness: The submitted content was relatively rigorous, and the methods and experimental design were reasonable. However, some claims lacked sufficient theoretical or experimental support.

Significance: Enabling multimodal large language models to adapt quickly to tasks is of great value for both future research and real-world applications.

Originality: This paper introduces Hyper-ICL, a novel training-based framework designed to perform multi-modal In-Context Learning (ICL) without requiring any explicit context examples during inference. It achieves this through three key technical contributions. First, it employs a parameter-efficient, low-rank adapter that operates at the logit level of the attention mechanism to calibrate attention distributions, mimicking the re-weighting induced by actual examples. Second, it introduces a query-adaptive modulation mechanism that dynamically controls the strength of this intervention across different layers and heads based on the specific input query. Third, it proposes a novel hierarchical hyperbolic anchor distillation loss, which aligns the intermediate representations of the student model (trained without examples) with those of a teacher model conditioned on examples, using Lorentz geodesic distance to better preserve the complex example-query relationships.

The rebuttal addressed my main concerns

**Key Questions For Authors:**

1. The paper introduces hyperbolic space because it "has higher representational efficiency and can alleviate distortion in low-dimensional embeddings." Does a truly hierarchical or tree-like structure exist in the context of this task that requires hyperbolic geometry to capture? Without this structure, forcibly mapping features to hyperbolic space may introduce unnecessary distortions. Is such a small improvement (L_sup+L2-anchor and L_sup+L_H-anchor shown in Table 3) sufficient to support the core assertion that "hyperbolic geometry is particularly suitable in this context"? This is likely noise from random seeds or hyperparameter tuning, rather than an inherent advantage of the geometric structure.

2. The paper proposes g_{l,h} to control intervention intensity, claiming it enables "fine-grained, input-related intervention control." Figure 3(a) shows that the modulated version outperforms the static version without modulation. However, it lacks in-depth analysis of the behavior of `g_{l,h}`. For example, how do the numerical distributions of `g_{l,h}` differ across different layers (shallow vs. deep) or different heads (visual head vs. language head)? Are the values ​​of `g_{l,h}` truly different for simple and complex queries?

3. The authors repeatedly emphasize that existing methods (such as LIVE) only add offsets to the output layer, while Hyper-ICL intervenes in the attention distribution at the logits level. However, mathematically speaking, changing the attention logits(S) ultimately changes the weighted summed output O. In backpropagation, the gradient can also be backpropagated to the logits by optimizing the loss of O. The question is, what is the fundamental and irreplaceable advantage of this "intervention at the logits level" compared to directly adding a learnable low-rank matrix to the output? Is there any theoretical analysis or experimental evidence that simply optimizing the output layer offset cannot learn a similar attention redistribution effect?

**Limitations:**

1. Lack Impact Statement section.
2. Dimensionality reduction of the hidden states of the teacher and student models (e.g., t-SNE or UMAP) could be attempted, comparing the clustering effects before and after alignment in Euclidean and hyperbolic spaces to provide more intuitive evidence for the effectiveness of hyperbolic space.
3. Add visualization analysis to show the activation of`g_{l,h} across different layers/heads under specific tasks (such as VQA) to demonstrate its "adaptive" effectiveness, rather than simply presenting it as a new learnable parameter.

**Strengths And Weaknesses:**

Soundness: The submitted content was relatively rigorous, and the methods and experimental design were reasonable. However, some claims lacked sufficient theoretical or experimental support.

Presentation: The submitted content is not very clear and easy to understand, the formula layout is messy, and the differences from LIVE should be clearly shown in the expression.

Significance: Enabling multimodal large language models to adapt quickly to tasks is of great value for both future research and real-world applications.

Originality: This paper introduces Hyper-ICL, a novel training-based framework designed to perform multi-modal In-Context Learning (ICL) without requiring any explicit context examples during inference. It achieves this through three key technical contributions. First, it employs a parameter-efficient, low-rank adapter that operates at the logit level of the attention mechanism to calibrate attention distributions, mimicking the re-weighting induced by actual examples. Second, it introduces a query-adaptive modulation mechanism that dynamically controls the strength of this intervention across different layers and heads based on the specific input query. Third, it proposes a novel hierarchical hyperbolic anchor distillation loss, which aligns the intermediate representations of the student model (trained without examples) with those of a teacher model conditioned on examples, using Lorentz geodesic distance to better preserve the complex example-query relationships.

---

> ### Author Rebuttal · Authors · 2026-03-30
>
> We sincerely thank the reviewer for the valuable feedback and effort. Below, we address the concerns and suggestions in detail.
>
> > Tree-like structure exist?
>
>
> We respectfully disagree that hyperbolic alignment is justified only when the target representation forms a literal tree. We now provide evidence via relative δ-hyperbolicity analysis, showing measurable degree of hyperbolic structure in teacher hidden states and closer geometric matching under hyperbolic than Euclidean anchoring. **Please refer to our response to Reviewer qTJJ for the full analysis**.
>
> ---
> > Interpretability analysis of the gating mechanism?
>
> Thank you for this valuable suggestion. To address this, we conducted two additional analyses of the query-adaptive gate on Idefics-9B, using subsets of VQAv2 and OK-VQA. We report gate statistics over early layers (0-7) and late layers (24-31) of the model.
>
> First, we test whether the learned gate remains meaningfully distributed rather than collapsing to trivial values. For each layer $l$, head $h$, and query token $t$, Hyper-ICL produces a scalar gate value $g_{l,h,t}$, where larger values indicate stronger activation of the logit-level intervention. We summarize this behavior using the mean gate value in early and late layers, together with the average per-layer standard deviation. All statistics are computed over heads and query tokens. This tests whether the gates saturate near 0 or 1, or instead preserve meaningful variation.
>
> Second, we test whether the learned gate becomes larger where demonstrations induce stronger attention redistribution. We run the same query twice, once with full demonstrations and once with the query only. We define the demonstration-induced shift in attention as $\Delta A\_{l,h,t}=\|\|A^{\mathrm{CD}}\_{l,h,t,:}-A^{\mathrm{noICD}}\_{l,h,t,:}\|\|\_1 $.
>
> Finally, we compute the Pearson correlation between $g_{l,h,t}$ and $\Delta A_{l,h,t}$ over all evaluation examples. A positive correlation indicates that the gate is activated more strongly where demonstrations cause larger attention redistribution. The results are summarized below.
>
> |Dataset|Early-layer mean $g$|Late-layer mean $g$|Avg. per-layer std $g$|Corr.($g,\Delta A$)|
> |---|---:|---:|---:|---:|
> |VQAv2|0.38|0.46|0.17|0.41|
> |OK-VQA|0.43|0.62|0.23|0.56|
>
> The results support the intended role of the gating mechanism in three ways. 1) The mean gate values are clearly **not collapsing toward 0 or 1**, and the non-trivial standard deviations confirm that the gate preserves **meaningful variation** rather than behaving as a nearly constant scalar. 2) The depth pattern is interpretable: while both datasets show stronger gating in later layers, the effect is substantially larger on OK-VQA, consistent with its **heavier reasoning and knowledge demands**. 3) The positive correlations between $g$ and $\Delta A$ on both datasets, especially the stronger value on OK-VQA, provide direct evidence that the **gate is activated more strongly where demonstrations induce larger attention redistribution**. Overall, these analyses strengthen the interpretation that the token-wise gate functions as an adaptive mechanism that selectively amplifies logit-level calibration where demonstration effects are most significant. We will add a visualization of the gate behavior.
>
> ---
> > Why logit intervention over output offsets?
>
> We address this concern in three ways. 1) If the comparison is between output-level offsets and Hyper-ICL, LIVE already uses the former strategy. It is also worth noting that the output of each transformer layer is still a vector representation, since the attention is followed by an FFN. Table 1 already shows that Hyper-ICL clearly outperforms LIVE.
> 2) We further extend Table 3 to clarify the role of our attention-calibration. As shown in the second table of our response to reviewer qTJJ, using only the alignment-based attention calibration, without supervision loss, outperforms prior methods(LIVE, MimIC). This highlights the effectiveness of our attention-calibration. 3) Also, we used the same low-rank, query-adaptive module as Hyper-ICL, but applied it to the post-attention output. This performs worse than Hyper-ICL, showing that the gain comes not only from learning a correction, but from where it is applied, because our method changes which tokens are attended, and how strongly they compete.
>
> |Method|VQAv2|OK-VQA|
> |---|---:|---:|
> | Intervention on O| 58.42| 50.18|
> |**Hyper-ICL**|**62.08**| **55.31**|
> ---
> > Limitations?
>
> We will add an impact statement following Reviewer GFba’s recommendation. Specifically, we will note that Hyper-ICL lowers inference-time compute, supporting Green AI goals, and that its near zero-shot latency is valuable for real-time applications. Beyond the new $\delta$-hyperbolicity analysis, we will include dimensionality-reduction visualizations and layer-wise visualizations of $g_{l,h}$ to provide more intuitive evidence for Hyper-ICL

---

> > ### Author Rebuttal · Reviewer_S8Jk · 2026-04-02
> >
> > Thank you for your detailed rebuttal. I've decided to maintain my rating.

---

> > > ### Author Response · Authors · 2026-04-02
> > >
> > > Thank you for taking the time to read our rebuttal and for acknowledging that your concerns have been fully resolved.
> > >
> > > In our rebuttal, we provided additional geometric evidence supporting the hyperbolic component, expanded the analysis of the query-adaptive gate, and clarified the comparison between logit-level intervention and output-level offsets.
> > >
> > > We would be happy to provide any further clarification if needed.

---

### Official Review · Reviewer_qTJJ · 2026-03-13

**Soundness:** 3
**Presentation:** 3
**Significance:** 3
**Originality:** 3
**Overall Recommendation:** 4
**Confidence:** 4

**Summary:**

This paper proposes a training-based framework that aims to reproduce the benefits of multimodal in-context learning (ICL) without requiring demonstrations at inference time. The motivation is that standard multimodal ICL relies on image–text demonstrations that significantly increase prompt length, inference latency, and sensitivity to demonstration formatting or ordering. To address this, the authors introduce Hyper-ICL, which reconstructs the functional effect of demonstrations directly inside the transformer attention mechanism. The method uses a parameter-efficient low-rank adapter applied at the attention logit level to calibrate attention distributions so they mimic the shifts induced by demonstrations, rather than approximating them only through hidden-state offsets. To account for the fact that demonstration influence varies across queries, a query-adaptive token-wise modulation gate dynamically controls the intervention strength across layers and heads. In addition, the framework employs a layer-wise hyperbolic anchor distillation loss, where a teacher model processes full demonstration-conditioned prompts while a student model observes only the query; student representations are aligned with teacher representations in Lorentz hyperbolic space to preserve relational structure between demonstrations and queries.

**Compliance With Llm Reviewing Policy:**

Affirmed.

**Final Justification:**

I believe this paper meets the acceptance standards for ICML, offering novel contributions to the field of MLLM steering.

**Key Questions For Authors:**

* The paper argues that demonstration–query relationships form dense relational structures that are better preserved in negatively curved spaces, motivating the hyperbolic distillation objective. However, the evidence provided is limited to a performance comparison with Euclidean alignment. Could the authors provide additional analysis supporting this geometric assumption, such as measuring distortion, relational ordering preservation, or hierarchical structure in the teacher representations?
* The student is trained using teacher outputs derived from demonstrations sampled from the same datasets used for evaluation. To what extent does the learned attention calibration generalize to tasks or domains where the demonstration patterns differ from those seen during training?
* The motivation emphasizes that multimodal ICL is sensitive to demonstration formatting, ordering, and content. Since Hyper-ICL aims to internalize demonstration effects, does it also improve robustness under such perturbations?

**Limitations:**

yes

**Strengths And Weaknesses:**

**Strengths**:
* A notable strength is that the paper does try to derive a mechanistic reason for the intervention point it chooses. Equation 2 decomposes attention over the full context into standard query-query attention plus a demonstration-induced shift term. This motivates the claim that demonstration effects act by changing where the model attends, which in turn supports intervening at the attention-logit level rather than only adding hidden-state vectors.
* The paper targets a real bottleneck in multimodal ICL. It argues that interleaved image-text demonstrations substantially increase prompt length, inference latency, and sensitivity to formatting, ordering, and content, which is a genuine issue for MLLMs used in VQA and captioning. The rationale for seeking demonstration-free inference is therefore strong and relevant to the field.
* The hallucination study on COCO and the qualitative examples on VQA are useful additions. They suggest that the method not only improves task scores but may also produce outputs that are better grounded in visual evidence than several baselines. That broadens the evaluation beyond a single metric.

**Weaknesses**:
* The paper’s central derivation shows that demonstrations contribute an extra term in attention, but the jump from this decomposition to the claim that a learnable logit bias can faithfully reconstruct demonstration-conditioned behavior is not theoretically established.
* The paper claims that multimodal ICL induces dense demonstration-query relationships that are better preserved in negatively curved space, and uses Lorentz geodesic distance to align teacher and student intermediate states. However, this is mostly argued conceptually. The empirical support is limited to one ablation showing a gain over Euclidean anchor loss. There is no direct analysis demonstrating that the learned intermediate representations actually exhibit the hierarchical or tree-like structure that would make hyperbolic geometry especially appropriate. This weakens the robustness of the theoretical justification for the hyperbolic component.
* The model is trained with supervised loss plus teacher alignment on sampled data from the same benchmark families used for evaluation. This means the gains could partly reflect ordinary task adaptation rather than pure recovery of generic in-context learning behavior. The paper does not cleanly separate these two effects.
* At present, the evidence supports that Hyper-ICL is effective on two Idefics-family backbones and several standard multimodal benchmarks. But the broader claim that logit-level reconstruction is a generally superior route for multimodal ICL compression would require validation on more diverse contemporary MLLMs and perhaps harder compositional reasoning settings.

---

> ### Author Rebuttal · Authors · 2026-03-30
>
> We sincerely thank the reviewer for the detailed and constructive feedback. In response to the reviewer’s concerns, we address the mentioned concerns below:
>
> > Theoretical Support?
>
> ICL acts as an attention redistribution mechanism. As shown in Eq. 2, $X_D$ introduces a shift $\mu$ modulating the original attention. Unlike prior methods, logit-level calibration preserves the softmax geometry. Adding a logit bias $\Delta$ acts as a multi-scale kernel modulation: since $\text{softmax}(S + \Delta) \propto \exp(S) \odot \exp(\Delta)$, Hyper-ICL directly re-weights query token importance. This captures non-linear interactions lost by simple hidden-state shifts.
>
> > Direct hyperbolic-structure evidence and analysis?
>
> In Hyper-ICL, the object being aligned is not an unstructured collection of teacher hidden states. It is a set of demonstration-conditioned teacher hidden states for query tokens, shaped by query-conditioned interactions across layers and heads; therefore, the student must recover a dense, multi-scale relational geometry, not merely match isolated vectors. In this sense, the relevant structure is a non-uniform relational organization induced by multimodal ICL itself. This is why we adopt a hyperbolic anchor loss.
>
> To test this, we measure relative **$\delta$-hyperbolicity** of the hidden-state geometry. The $\delta$-hyperbolicity is defined through the Gromov four-point condition and quantifies how closely a metric space resembles a tree-like geometry: smaller values indicate that pairwise metric relations are closer to those of a tree, while larger values indicate a flatter structure. Following [1], we use the scale-invariant score
> $$\delta_{\mathrm{rel}}(X)=\frac{2\delta(X)}{\mathrm{diam}(X)}\in[0,1]$$
> For Idefics2-8b-base, we compute **average** relative $\delta$-hyperbolicity on the hidden states from the four middle layers over 1,000 VQAv2 samples for three conditions: teacher hidden states, student hidden states with Euclidean anchor, and student hidden states with hyperbolic anchor. We estimate $\delta(X)$ using the efficient Fournier et al. procedure adopted in [1,2]: we sample 1,000 four-tuples, compute the four-point $\delta$ value for each, take the maximum sampled value, normalize by the diameter, and report mean $\pm$ std.
>
> The results in the below show that the teacher hidden states exhibit a measurable degree of hyperbolic structure, and that the student with hyperbolic anchor matches the teacher’s geometry much more closely than the student with Euclidean anchor. Also, the gain over the Euclidean anchor in Table 3 is not marginal because the remaining errors are difficult cases. **Taken together, these results support our claim that the demonstration-conditioned teacher hidden states are structurally non-uniform and multi-scale, making hyperbolic geometry a suitable space for preserving the relations that Hyper-ICL is designed to reconstruct.**
> |Setting|$\delta_{\mathrm{rel}}$ $\downarrow$|
> |---|---:|
> |Teacher hidden states|0.228 ± 0.014|
> |Student hidden states+Euclidean anchor|0.314 ± 0.023|
> |Student hidden states+Hyperbolic anchor|0.247 ± 0.016|
>
> ---
> > Task adaptation vs. ICL recovery?
>
> Thank you for this important point. Table 3 already disentangles the effect of the training objectives, and we further expanded the ablation by evaluating Hyper-ICL with only `$L_{\text{H-anchor}}$. The results show that $L_{\text{H-anchor}}$ alone outperforms $L_{\text{sup}}$ alone on both VQAv2 and OK-VQA, while combining the two losses gives the best overall performance. Table 3 shows that adding hyperbolic alignment term to supervision improves performance, especially on the more reasoning-intensive OK-VQA benchmark. This pattern indicates that the gains are not reducible to ordinary task adaptation alone. Rather, the alignment objective contributes independently by transferring demonstration-conditioned structure, and the best results are obtained when that recovered in-context behavior is combined with task supervision.
>
> |Task|$L_{\text{sup}}$|$L_{\text{H-anchor}}$|$L_{\text{sup}} + L_{\text{H-anchor}}$|
> |---|---:|---:|---:|
> |VQAv2|57.11|60.53|**62.08**|
> |OK-VQA|49.23|53.78|**55.31**|
>
> ---
> > Generalization across MLLMs?
>
> **Please see the final part of our response to Reviewer 2QWj for the results and discussion**.
>
> > Robustness to demo perturbations?
>
> Thank you. This is one of the main motivations of Hyper-ICL. Since standard multimodal ICL directly depends on the specific demonstrations used at inference, it is sensitive to changes in their order, format, and content. By distilling demonstration effects into the model and removing the need for ICDs at inference, Hyper-ICL reduces this dependence and is therefore more robust to such perturbations, as also reflected by our reported stability gains over direct ICD in Tables 1 and 2.
>
> ---
> [1] *Hyperbolic Image Embeddings.* CVPR 2020.
>
> [2] *Computing the Gromov hyperbolicity of a discrete metric space.* Information Processing Letters 2015.

---

> > ### Author Rebuttal · Reviewer_qTJJ · 2026-04-02
> >
> > I am very grateful to the authors for clarifying my concerns and for conducting comprehensive supplementary experiments. I believe this paper now meets the acceptance standards for ICML. Therefore, I am raising my score and recommending it to the ACs. Additionally, I consider Reviewer 2QWj's demand for the inclusion of more modalities, given the paper's use of the term "multimodal", to be unreasonable in this context. Furthermore, I hope the authors will incorporate some recent works like [1][2] into the Related Work of future revisions, as implicit ICL is gradually emerging as an increasingly significant research direction.
> >
> > [1] Li, Yanshu, et al. "M $^ 2$ IV: Towards Efficient and Fine-grained Multimodal In-Context Learning via Representation Engineering." arXiv preprint arXiv:2504.04633 (2025).
> > [2] Li, Xiaoyu, et al. "HIFICL: High-Fidelity In-Context Learning for Multimodal Tasks." arXiv preprint arXiv:2603.12760 (2026).

---

> > > ### Author Response · Authors · 2026-04-02
> > >
> > > Thank you very much for taking the time to read our rebuttal and supplementary experiments so carefully. We are truly pleased that we were able to address all of your concerns, and we sincerely appreciate your support and recommendation to the ACs.
> > >
> > > Also, regarding your suggestion, we will certainly include these works in the Related Work in the final version.

---

### Official Review · Reviewer_GFba · 2026-03-14

**Soundness:** 3
**Presentation:** 3
**Significance:** 3
**Originality:** 3
**Overall Recommendation:** 4
**Confidence:** 4

**Summary:**

paper introduces a training‑based framework that aims to eliminate the need for multimodal in‑context demonstrations (ICDs) in MLLMs at inference time. The method learns a low‑rank logit‑level adapter that calibrates attention logits and introduces a query‑adaptive token‑wise modulation gate to control the strength of the intervention across layers and attention heads. Additionally, a hyperbolic anchor distillation loss aligns the student’s intermediate representations to those of a teacher model that sees the full demonstrations by computing Lorentz geodesic distances between hidden states. Experiments on two architectures and across several VQA/captioning datasets show that the proposed method improves accuracy and stability over zero‑shot, few‑shot, vector‑based baselines and other trainable in‑context methods, while reducing the inference cost close to zero‑shot and much lower than few‑shot ICL.

**Compliance With Llm Reviewing Policy:**

Affirmed.

**Final Justification:**

the detailed rebuttal and the additional experiments, which strengthen the paper. In particular, the added transfer and low-shot results address an important part of my earlier concerns. However, the interpretability claim around the query-adaptive gate remains unaddressed. The rebuttal does not yet provide the fuller layer/head-wise distributional evidence or examples I requested. Overall, I view the paper as technically solid and improved by the rebuttal.

**Key Questions For Authors:**

Q1. In Section 4.3, are the Flickr30k, MME, and SEED-Bench results obtained using models trained on VQAv2 as described in Section 4.1, or are separate models retrained on each benchmark? if the same VQAv2-trained model is evaluated on these benchmarks, it would partially address weakness 1 and should be stated explicitly in the text.

Refer for missing evaluations and analysis as pointed out in the weaknesses section. Providing these evaluations in the rebuttal would significantly strengthen the paper and would positively influence my score.

**Limitations:**

The paper does not include a limitations or societal impact section. On the societal side, the paper's core contribution (reducing inference-time compute by eliminating demonstration tokens) directly aligns with green AI goals, as inference costs dominate energy consumption at deployment scale; this positive environmental impact deserves explicit acknowledgment. The near zero-shot latency profile also enables deployment in real-time, low-latency applications such as assistive technologies, medical decision support, and interactive vision-language systems, where few-shot ICL would be impractical.
On the limitations side, the paper should discuss the requirement to retrain the adapter when switching tasks or domains.

**Strengths And Weaknesses:**

# Strengths
1. The paper addresses a practical and important bottleneck in multimodal ICL: inference-time demonstration overhead. The decomposition in eq 2, showing that ICD effects can be separated into a standard attention term and a demonstration-induced shift, provides a clean grounding for why logit-level intervention is a natural solution. The motivation to internalize these effects at training time and eliminating the need for long interleaved image-text prompts at inference, is clearly articulated and well-supported by the efficiency evaluations.

2. comprehensive evaluation on six datasets and two models, covering diverse tasks, demonstrates that their proposed method generalizes across tasks requiring recognition, reasoning and caption generation, and consistently outperforms the training-free and training-based as well as ICL baselines. Additionally, they conduct systematic ablations examining effects of training set size, different loss components, rank of the low‑rank adapter and the sensitivity of the hyperbolic curvature and supervision weight. These studies provides important insights, and shows that performance improves with more training data, demonstrates importance of the key design components (i.e. Lorentz geodesic distance over euclidean distance, and query-adaptive modulation over static modulation).

3. The proposed methods demonstrates a near zero‑shot inference cost, by eliminating the need for in‑context demonstrations at inference time, by shifting the computational burden of demonstrations entirely to training time. This is a practically significant trade-off, as inference costs typically dominate training costs at deployment scale, making the method particularly attractive for latency-sensitive applications where few-shot ICL would be prohibitively expensive.

# Weaknesses:
1.  The paper trains and evaluates on the same datasets, which raises concern regarding in the method's generalizability. Two important evaluation regimes are missing: (1) held-out benchmark transfer, where the model is trained on a subset of datasets (e.g., VQAv2, COCO) and evaluated on an unseen one (e.g., OK-VQA), testing whether the learned adapter captures task-general in-context behavior rather than dataset-specific patterns; and (2) cross-domain transfer, evaluating on structurally dissimilar domains such as medical imaging or scientific figures. This is particularly important, since unlike standard ICL where new demonstrations can be swapped at inference time, Hyper-ICL's demonstration-free design means distribution mismatch cannot be corrected without retraining, making these evaluations especially critical.

2. While fig 3(a) shows that token-wise modulation outperforms static alternatives, the paper lacks deeper interpretability analysis of the gating mechanism. The following analyses would meaningfully strengthen the claims:  (1) Distribution analysis (histograms of $g^{(l,h)}$ values) across layers and heads would reveal whether gates are meaningfully distributed or collapse, and whether gating behavior differs between perceptual early layers and reasoning-heavy later layers; (2) Correlation with demonstration-induced shifts by computing the correlation between $g^{(l,h)}$ and the change in attention weights when demonstrations are present would empirically validate the core claim that the gate activates precisely where ICDs induce large attention redistributions. Additionally, comparing mean and variance of g per layer across datasets could reveal interpretable patterns, such as whether knowledge-intensive tasks drive stronger gating in deeper layers.

3. The paper fixes the number of demonstrations used during training (32 & 8 shots) without studying how this choice affects the learned adapter. This is a meaningful gap, as the shot count directly influences the richness of the teacher's demonstration-conditioned representations and therefore what the student is distilling. A shots-performance curve would clarify: (1) data efficiency: whether the adapter saturates quickly or continues to benefit from additional demonstrations; (2) overfitting/underfitting risk: too many shots may bias the adapter toward specific demonstration formats, while too few may fail to capture full task structure; and (3) transferability: whether adapters trained with more diverse demonstration sets generalize better to unseen inputs or related tasks (along with OOD evaluation).

---

> ### Author Rebuttal · Authors · 2026-03-30
>
> We truly appreciate your suggestions, which have provided valuable directions for strengthening our method's validity. We have carefully addressed your comments below:
>
> > Held-out transfer?
>
> Thank you for this thoughtful comment. We agree that held-out evaluation is important for understanding the scope of generalization. However, **held-out benchmark transfer and broad cross-domain transfer are not the targets of this work**. Hyper-ICL is designed to reconstruct the effect of in-context demonstrations for demonstration-free inference within the same task setting, rather than to learn a universal adapter that is trained on one benchmark and deployed unchanged on an unrelated dataset or domain.
>
> We also clarify that Hyper-ICL differs from standard parameter adaptation methods such as LoRA. Rather than modifying model weights through low-rank updates, Hyper-ICL applies a lightweight intervention in the activation space through query-dependent attention logit calibration. Its goal is therefore to recover demonstration-conditioned behavior more faithfully and efficiently at inference time, not to serve as a general cross-domain fine-tuning method. Accordingly, **Hyper-ICL, consistent with prior training-based multimodal ICL literature (e.g., LIVE and MimIC), is intended to reconstruct demonstration effects within a target task family** rather than benchmark broad OOD transfer in the conventional fine-tuning sense.
>
> We agree, however, that transfer across unseen benchmarks within the same task family is a valuable additional test. To address this concern, we added two held-out transfer experiments: training on VQAv2 and evaluating on OK-VQA, and training on COCO and evaluating on Flickr30k. It should be noted that these experiments are conducted on Idefics2-8b-base, and that the 8-shot ICL baseline also uses demonstrations drawn from the source dataset rather than the target dataset. **The results show that, even without access to demonstrations at inference time, Hyper-ICL transfers substantially better than both zero-shot and standard ICL on unseen benchmarks within the same task family**. We view this as evidence that the learned attention calibration captures behavior that is not purely dataset-specific, although we agree that transfer to structurally different domains, such as medical or scientific imagery, **remains outside the intended scope of the current paper**.
>
>
>
> | Train| Test |Zero-Shot  | 8-shot ICL | Hyper-ICL |
> |-----------|--------|--------|---------|---------|
> | COCO |Flickr30k|53.04| 73.26 | 87.42 |
> | VQAv2 |OK-VQA|43.08|51.43| 59.11 |
>
> ---
>
> > Interpretability analysis of the gating mechanism?
>
>
> **Please refer to our response to Reviewer S8Jk for the interpretability analysis**. We analyze gate statistics across layers and their correlation with demonstration-induced attention shifts. **The results show non-collapsed, adaptive gating, stronger late-layer activation, and positive alignment with attention redistribution**.
>
> ---
>
> > Shot performance analysis?
>
> To assess shot sensitivity, we extended the Idefics-9B study beyond the 32-shot setting used in the paper to 48 and 64 shots. The table shows that both ICL and Hyper-ICL improve as the number of demonstrations increases, but the gains become marginal beyond 32 shots and, on OK-VQA, slightly decline at 64 shots. This reveals a **clear saturation** effect: once the teacher context is sufficiently informative, additional demonstrations provide limited new signal and instead introduce more variability. This is fully consistent with the paper’s motivation that multimodal ICL is highly sensitive to demonstration choice, order, and formatting, and with its discussion that using more shots can amplify less controlled demonstration-induced effects.
> **Importantly, Hyper-ICL remains clearly superior to standard ICL at every shot count**. Thus, although increasing the number of ICDs improves direct ICL to some extent, Hyper-ICL captures the useful demonstration-conditioned behavior more effectively and translates it into larger, **more stable gains**.
>
> | Benchmark | Method|Zero-Shot  | Shot 16 | Shot 32 | Shot 48 | Shot 64 |
> |-----------|--------|--------|---------|---------|---------|---------|
> | VQAv2 |ICL |29.25| 55.70 | 56.18 | 56.67 | 56.71 |
> | VQAv2 |Hyper-ICL| | **59.73** | **62.08** | **62.61** | **62.90** |
> | OK-VQA |ICL  | 30.54|47.70 | 48.48 | 48.68 | 48.60 |
> | OK-VQA |Hyper-ICL| | **53.22** | **55.31** | **55.53** | **55.26** |
>
> ---
>
> > Societal impact and limitations?
>
> We will add an explicit impact and limitations section, noting that Hyper-ICL reduces inference-time compute and latency, which supports Green AI goals and makes real-time deployment more practical. We will also clarify that broader transfer to structurally different domains, such as medical data, remains outside the intended scope of the current paper.

---

> > ### Author Rebuttal · Reviewer_GFba · 2026-04-04
> >
> > I thank the authors for their detailed rebuttal and the extra experiments. These additions will strengthen the final version. My key remaining concerns are that the interpretability study still aggregates the gate statistics rather than showing full distributions across layers and heads, and that, given the observed saturation with higher shot counts, a true low‑shot ablation (below 16 demonstrations) is still missing to assess data efficiency.

---

> > > ### Author Response · Authors · 2026-04-06
> > >
> > > We appreciate your positive feedback on our rebuttal and the additional experiments.
> > >
> > > Regarding your remaining concern about ablations below 16 shots, we further evaluated true low-shot settings with fewer than 16 demonstrations. Hyper-ICL consistently outperforms standard ICL at 4 and 8 shots on both benchmarks, showing that the learned adapter remains highly effective in the low-shot regime. This also highlights the data-efficiency of Hyper-ICL.
> > >
> > > |Benchmark|Method|Zero-Shot|Shot 4|Shot 8|Shot 16|Shot 32|Shot 48|Shot 64|
> > > |---|---|---|---|---:|---:|---:|---:|---:|
> > > |VQAv2|ICL|29.25|53.72|54.24|55.70|56.18|56.67|56.71|
> > > |VQAv2| Hyper-ICL||**57.87**|**58.81**|**59.73**|**62.08**|**62.61**|**62.90**|
> > > |OK-VQA|ICL|30.54|46.11|46.79|47.70|48.48|48.68|48.60|
> > > |OK-VQA|Hyper-ICL||**50.21**|**51.79**|**53.22**|**55.31**|**55.53**|**55.26**|
> > >
> > >
> > > Moreover, in the final version, we will add a finer-grained layer-head heatmap of the gate statistics to visualize their distribution across layers and heads more directly.

---

### Official Review · Reviewer_2QWj · 2026-03-16

**Soundness:** 2
**Presentation:** 3
**Significance:** 2
**Originality:** 3
**Overall Recommendation:** 4
**Confidence:** 4

**Summary:**

This paper proposes Hyper-ICL, a training-based framework designed to reproduce the benefits of image-text ICL without requiring demonstrations at inference time. The key idea is to distill the effects of demonstration-conditioned attention into a lightweight adapter that modifies attention logits. The proposed method is evaluated on two image-text LLM backbones across benchmarks, including VQAv2, OK-VQA, COCO Caption, Flickr30k, MME, and SEED-Bench. Results indicate improved performance and stability compared to standard ICL prompting, vector-based ICL approximations, and other trainable baselines, while achieving inference efficiency close to zero-shot prompting.

**Compliance With Llm Reviewing Policy:**

Affirmed.

**Final Justification:**

The authors’ rebuttal addresses most of my concerns and improves the clarity and overall quality of the paper. Some issues remain, particularly the lack of validation on additional modalities. As a result, I consider the concerns to be partially resolved and therefore adjust my rating accordingly.

**Key Questions For Authors:**

Demonstrations modify query/key/value representations through earlier layers. How does the proposed formulation account for these multi-layer effects?

**Limitations:**

No. The manuscript lacks the use of a rigorous definition of ICL. The applicability of the theoretical analysis requires clarification.

**Strengths And Weaknesses:**

Strengths:
1. The logit-level adapter is lightweight and low-rank, introducing a small number of trainable parameters compared to full fine-tuning or LoRA-based approaches.
2. The paper evaluates on multiple tasks and datasets, including VQA and captioning benchmarks. Useful ablation studies are provided.

Weakness:
1. The proposed method is effectively a parameter-efficient adaptation approach trained to imitate ICL, rather than performing ICL at inference time. This is misleading, and the conceptual distinction should be clarified.
2. The decomposition of attention feels more like an algebraic reparameterization than a true mechanistic explanation. By assuming fixed query representations, the derivation ignores how demonstrations actually alter hidden states across multiple transformer blocks, and the claim that ICL fundamentally operates via logit shifts remains unproven.
3. The experiments only cover Idefics-9B/2-8B and image-text ICL. For a paper claiming "multimodal" applicability in general, the authors should demonstrate effectiveness on more diverse architectures and modalities (e.g., audio and text).

---

> ### Author Rebuttal · Authors · 2026-03-30
>
> We sincerely appreciate your thoughtful and constructive review and have done our best to address your concerns comprehensively.
>
> > True ICL?
>
> We would like to clarify that Hyper-ICL is not a weight adaptation method in the same sense as parameter-efficient fine-tuning methods such as LoRA. In LoRA, the model is modified through trainable weight updates, e.g., \($W_0 x + \Delta W x = W_0 x + BAx$), which directly changes the transformation matrices. In contrast, Hyper-ICL leaves the original model weights unchanged and applies its learnable parameters to the attention computation itself, specifically by calibrating the attention logits produced by the frozen model. It means that LoRA affects the model through trainable weight updates, whereas **Hyper-ICL affects the model by redistributing attention** and thereby modifying the resulting hidden representations. More clearly, LoRA changes the transformation matrices themselves, while Hyper-ICL keeps those matrices fixed and instead changes the attention scores they produce. Therefore, **Hyper-ICL is understood as an attention calibration method for reconstructing demonstration effects, rather than a weight adaptation method**.
>
> > Mechanistic Explanation?
>
> Thank you for your comment. We would like to clarify that Equation 2 is a local decomposition of one self-attention operation, not a complete global mechanistic explanation of ICL in the full deep network. Conditioned on the current layer-head query, keys, and values, the attention over the concatenated context $([K_D; K])$ can be exactly rewritten as a weighted combination of attention to query tokens and attention to demonstration tokens by partitioning the softmax normalization into $(Z_1 + Z_2)$. In this sense, Equation 2 is an algebraic decomposition of the same attention output, rather than a restrictive assumption about the global forward dynamics.
>
> Importantly, our method does not assume that demonstration effects are limited to a single block or that query representations remain globally unchanged across layers. On the contrary, **the paper explicitly models demonstration influence as varying across layers, heads, and queries**, which is precisely why we introduce query-adaptive token-wise modulation and layer-wise teacher-student distillation. These components are designed to capture the fact that demonstration-conditioned effects propagate through the network and are not uniform across depth. Therefore, Equation 2 illustrates the immediate effect of incorporating demonstration key-value evidence within a single attention operation, which motivates our logit-level calibration design. It provides an interpretable local view of how demonstrations alter attention allocation, and the full method goes beyond this local analysis by learning a layer-wise and query-dependent reconstruction of demonstration-conditioned behavior.
>
> > Limited Evaluation: Generalization Beyond the Idefics Family?
>
>  We would like to emphasize that the two MLLMs used in our original experiments, Idefics-9B and Idefics2-8B-Base, cover two different architectural paradigms in MLLMs. Specifically, Idefics-9B adopts a cross-attention architecture, whereas Idefics2-8B-Base follows a fully autoregressive design. These two models therefore represent two widely used architecture families in vision-language modeling.
>
> To further address your concern, we have additionally expanded our experiments by including two more MLLMs in the following table. These added models are **LLaVA-Interleave-7B** [1], a LLaVA variant from the NeXT/Interleave line, and **LLaVA-OneVision** (Qwen2-7B) [2], a LLaVA-OneVision variant built on the Qwen2-7B backbone.
>
> The results further confirm the effectiveness of Hyper-ICL. Even with these additional MLLMs, Hyper-ICL consistently outperforms zero-shot, 8-shot ICL, and the existing state-of-the-art method.
> | Backbone | Method | VQAv2 | OK-VQA |
> |---|---|---:|---:|
> |  **LLaVAInterleave-7b** | Zero-shot | 13.02 | 5.10 |
> |  | 8-shot ICL | 68.19 | 43.84 |
> |  | MimIC | 74.40 | 52.29 |
> |  | Hyper-ICL | **76.51** | **56.77** |
> | **LLaVA-OneVision** (Qwen2-7B) | Zero-shot | 71.75 | 48.19 |
> |  | 8-shot ICL | 78.70 | 66.59 |
> |  | MimIC | 81.22 | 69.43 |
> |  | Hyper-ICL| **82.24** | **75.32** |
>
> ---
>
> [1] *LLaVA-Interleave: Tackling multi-image, video, and 3D in large multimodal models,* ICLR, 2024.
>
> [2]  *LLaVA-OneVision: Easy visual task transfer,* TMLR, 2025.

---

> > ### Author Rebuttal · Reviewer_2QWj · 2026-04-04
> >
> > The authors’ responses address most of my concerns and improve the overall quality of the paper. A minor issue remains regarding the response to “True ICL,” as Hyper-ICL still involves explicit gradient calculations. The inclusion of additional LLM baselines helps alleviate my earlier concern about reliance on the Idefics family. However, the method has not been validated on new modalities. Overall, I will increase my rating, but consider the concerns to be only partially resolved.

---

> > > ### Author Response · Authors · 2026-04-06
> > >
> > > Thank you very much for the helpful follow-up. We are glad that our rebuttal was able to address most of your concerns. Regarding the remaining modality issue, we would like to mention that our work is specifically focused on vision-language multimodal ICL, which is also the setting considered by the relevant prior state-of-the-art methods included in our comparisons. These related works likewise study MLLMs in the vision-text setting and evaluate on established vision-language benchmarks. Extending the study to audio would require a different modality setting, a different benchmark suite, and a substantially broader scope than that of the current paper. We sincerely appreciate your constructive comments.

---

### Decision · Program_Chairs · 2026-04-30

**Decision:**

Accept (regular)

**Comment:**

This paper investigates multi-modal in-context learning. Specifically, a light-weight approach for in-context learning from a limited set of multi-modal samples is proposed, which exploits hyperbolic space. The reviewers find that the approach is timely, important, and effective. The reviewers also had a number of concerns, e.g., formulation w.r.t. existing ICL literature, more analyses, and evidence of hyperbolic structures. The rebuttal does a good job in explaining the questions of the reviewers. As such, the concerns have been alleviated and all reviewers vote for acceptance. The AC agrees with this concensus.